# Validated Ultrasound Speckle Tracking Method for Measuring Strains of Knee Collateral Ligaments In-Situ during Varus/Valgus Loading

**DOI:** 10.3390/s21051895

**Published:** 2021-03-08

**Authors:** Félix Dandois, Orçun Taylan, Johan Bellemans, Jan D’hooge, Hilde Vandenneucker, Laura Slane, Lennart Scheys

**Affiliations:** 1Development and Regeneration Department, Institute for Orthopaedic Research and Training (IORT), KU Leuven, 3000 Leuven, Belgium; orcun.taylan@kuleuven.be (O.T.); hilde.vandenneucker@uzleuven.be (H.V.); lennart.scheys@kuleuven.be (L.S.); 2Department of Orthopaedics, Ziekenhuis Oost Limburg (ZOL), 3600 Genk, Belgium; johan.bellemans@zol.be; 3Cardiovascular Imaging and Dynamics, Cardiovascular Sciences Department, KU Leuven, 3000 Leuven, Belgium; jan.dhooge@uzleuven.be; 4Department of Orthopaedics, University Hospitals Leuven, 49 Herestraat, 3000 Leuven, Belgium; 5Department of Mechanical Engineering, University of Rochester, Rochester, NY 14602, USA; lslane@ur.rochester.edu

**Keywords:** digital image correlation, knee collateral ligaments, speckle tracking, strains, ultrasound

## Abstract

Current ultrasound techniques face several challenges to measure strains when translated from large tendon to in-situ knee collateral ligament applications, despite the potential to reduce knee arthroplasty failures attributed to ligament imbalance. Therefore, we developed, optimized and validated an ultrasound speckle tracking method to assess the in-situ strains of the medial and lateral collateral ligaments. Nine cadaveric legs with total knee implants were submitted to varus/valgus loading and divided into two groups: “optimization” and “validation”. Reference strains were measured using digital image correlation technique, while ultrasound data were processed with a custom-built speckle tracking approach. Using specimens from the “optimization” group, several tracking parameters were tuned towards an optimized tracking performance. The parameters were ranked according to three comparative measures between the ultrasound-based and reference strains: R^2^, mean absolute error and strains differences at 40 N. Specimens from the “validation” group, processed with the optimal parameters, showed good correlations, along with small mean absolute differences, with correlation values above 0.99 and 0.89 and differences below 0.57% and 0.27% for the lateral and medial collateral ligaments, respectively. This study showed that ultrasound speckle tracking could assess knee collateral ligaments strains in situ and has the potential to be translated to clinics for knee arthroplasty-related procedures.

## 1. Introduction

Ultrasound (US) imaging is one of the most commonly used clinical modalities to assess the structural and geometric properties of soft tissues such as tendons and ligaments, though the soft-tissue biomechanical properties, e.g., tensile and viscoelastic properties, are at least equally important to obtain a thorough evaluation of their conditions [1]. Such biomechanical characterizations based on US speckle tracking have already been used to quantify tissue deformations of larger structures, such as the patellar and Achilles tendons [2,3,4]. However, these validated approaches are not directly applicable to quantify strains in smaller structures, such as the lateral collateral ligament (LCL) and medial collateral ligament (MCL), in situ without additional work [5], despite its immediate potential for clinical applications—specifically, for total knee arthroplasty (TKA) [5,6]. Although TKAs are common, with a volume of 1.5 million implants per year in countries part of the organization for economic cooperation and development alone and an expected four-fold increase by 2030 [7], TKA still fails and leads to revision surgery in 2–8% of cases [8]. Importantly, in more than 35% of cases, these failures are attributed to a ligament imbalance, leading to either excessive stiffness or instability of the postoperative joint [9]. Currently, however, ligament balancing procedures rely primarily on the subjective opinions of surgeons and/or assisting tools that collect indirect measurements of collateral ligament strains (e.g., intra-articular pressure distributions or bony distances) [10]. Therefore, since the collateral ligaments are two of the primary knee stabilizing structures post-TKA [11,12], an assessment of their in-situ biomechanical properties could be of great value. By providing surgeons with the currently unobtainable objective data, ligament balancing could be improved, thereby minimizing TKA failures attributed to ligament imbalance.

Prior studies have already explored the use of US speckle tracking to assess collateral ligament strains [5,6]. However, most studies were only validated ex situ on isolated LCLs [6] or, as previously highlighted [5], were lacking ground truth data. In the first study [6], the ex-situ validation could not consider the specific motion behavior of these structures in situ and the associated challenges [5] nor the presence of an implant, which could modify the acoustic impedance and further complicate tracking in the intended clinical application. In the second study [5], the lack of ground truth data during prior in-situ attempts to develop a US-based methodology prevented the optimization of the main processing parameters and a more thorough validation.

Nonetheless, these studies clearly illustrated the potential of US speckle tracking to measure the knee collateral ligament in-situ strains, even though its transfer from bench to bedside requires a more thorough validation. Therefore, the aim of this work is to present a dedicated US speckle tracking method for the in-situ evaluation of collateral ligament strains post-TKA, as well as the optimization of its main parameters and its validation based on the ground truth data collected using three-dimensional (3D) digital image correlation (DIC) [6,13,14].

## 2. Materials and Methods

### 2.1. Specimens Preparation

Following ethical approval (NH019 2017-02-03), a pragmatic sample of nine cadaveric knee joints, from five male specimens (age = 83.4 ± 6.69 years; body mass index (BMI) = 27.67 ± 2.77 kg/m^2^) with total knee implants were collected. Upon clinical inspection by experienced knee surgeons, two specimens showed ruptured LCLs. Consequently, a total of nine MCLs and seven LCLs could be investigated. First, bicortical bone pins were inserted into the tibia and the femur, on which rigid marker frames containing four reflective spheres each were attached, and computed tomography (CT) scans were acquired [15]. One day prior to testing, the legs were taken out of the freezer to thaw. Both the LCL and MCL were exposed, and the joint line was indicated on both ligaments for future referencing.

Legs were then positioned in a custom test bench maintaining a constant flexion–extension angle of 30°. Herein, the femur was rigidly attached to the test bench using bone pins restraining all degrees of freedom, and the tibia was free to move and rotate relative to the femur, being only constrained in terms of flexion by the bench itself (Figure 1).

### 2.2. Acquisitions

For all the acquisitions, a varus (LCL) or valgus (MCL) load ranging from 0 to at least 40 N was applied with a load cell (Series 4, Mark-10, Copiague, NY, USA) attached to the tibia at 35 cm from the joint line. Thus, real-time load feedback was provided to the operator, and the applied loads were recorded. During all trials, the relative trajectories of the reflective spheres attached to the tibia and the femur were tracked with a six-camera 3D motion capture system (MX40+, Vicon Motion Systems, Oxford, UK) operating at 100 Hz. In order to perform post-hoc comparisons between DIC and US trials over a matching loading range and kinematics, all signals—loaded cells, motion capture trigger and US trigger—were simultaneously collected and synchronized at 1000 Hz with LabView (National Instruments Corporation, Austin, TX, USA) for all acquisitions.

During the first series of loading cycles, US data were collected with a 38-mm, 10-MHz linear array transducer (Ultrasonix Corp., Richmond, BC, USA). The probe was aligned in parallel with the longitudinal direction of the ligament, centered on the joint line, and a stand-off gel pad was placed between the probe and the ligament. Then, the radiofrequency (RF) and b-mode data were collected dynamically, during loading cycles, at 70 frames per second, with a depth of 2 cm. For each US acquisition, the loading was cyclically repeated until a complete load range was recorded with acceptable image quality, as visually assessed in real time by the experienced ultrasonographer. For each specimen, the acquisition order was randomized between LCL and MCL, and three-to-five acquisitions with visually acceptable quality were obtained per ligament.

In a second series of loading cycles, DIC data were collected. For each DIC acquisition, three loading cycles were recorded in order to replicate the aforementioned cyclic loading conditions. Ligaments were prepared for DIC acquisition following similar methodology to the previous studies on soft tissues [6,14,16]. Diluted methylene blue was applied on the ligament to create background and contrast. Then, a speckle pattern was applied on the ligament with water-based white paint using an airbrush gun [17] (Figure 2). Afterwards, two high-resolution cameras (Flir Grasshoper3 GS3-U3-51S5M-C 5MP) with large field-of-view (FOV) lenses (Schneider Kreuznach Apo-Xenoplan 1.4/23) were placed, taking into account the optical properties of the lenses and the required FOV. Subsequently, the DIC system was calibrated for the acquisition using commercial software VIC3D (v8, Correlated Solutions, Inc., Columbia, SC, USA). Afterwards, data were recorded in triplicate at 20 Hz throughout the three loading cycles. Camera acquisitions were triggered and synchronized by the motion capture system’s trigger signal.

### 2.3. Processing

Prior to processing, specimens were divided into two groups. Five randomly selected specimens were appointed to the “optimization” group used to optimize the tracking parameters described further in this section for both the LCL and MCL. The four remaining specimens, including the two specimens with damaged LCLs, constituted the “validation” group, which were only processed with the unchanged optimal parameter sets obtained from the “optimization” group.

For both the DIC and US acquisitions, only data corresponding to the last load cycle of each trial were processed. Motion capture data, camera image sequences (DIC) and RF data (US) were cropped according to the synchronized load cell data (with a 5-ms accuracy) to match a loading range from 0 to 40 N.

#### 2.3.1. Ultrasound Processing

Ultrasound data were processed in MATLAB (MATLAB R2016a, The MathWorks, Inc., Natick, MA, USA) with a processing pipeline adapted from a method previously validated on the Achilles tendon [13,18]. This processing workflow was divided into three phases: pre-tracking, tracking and post-tracking.

In the first pre-tracking phase, the quality of the cropped US images was subjectively assessed a second time, and trials presenting a clear transducer motion, implant artefact or important out-of-plane motion were excluded (25% of the discarded trials for both LCLs and MCLs). For good quality trials, the nodes were defined on the ligament’s b-mode image around the joint line, using the lateral (LCL) or medial (MCL) femoral condyles as anatomical references. The superficial and deep borders of the ligament were manually outlined. Then, equidistant nodes were automatically placed along a line of 18 mm of length positioned at 50% of the ligament depth, with a 0.8-mm inter-node distance (Figure 3). Two additional points were added to allow tracking the superficial border of the ligament. Afterwards, RF data were upsampled by a factor two and four along the image x- and y-axis respectively, to increase the spatial density for further correlation analysis [19].

In the second phase, i.e., the tracking phase, a speckle tracking approach was implemented to measure displacements within the ligaments (Figure 4). For each frame, a to-be optimized kernel was centered on each node, and a to-be optimized search window was centered on the same node position in the subsequent frame. Then, the maximum of the normalized cross-correlation between the kernels and the search windows was used to compute the gross displacements. Afterwards, for the nodes with a maximum correlation coefficient above a to-be optimized threshold, subpixel displacements were computed to improve the accuracy by fitting a quartic spline around the gross displacements (5 pixels in both directions) and finding their maximum. For the nodes with a maximum correlation coefficient below the threshold, spatial interpolation was used to estimate their displacement based on the neighbor node displacements. Then, all results were median filtered, and displacements were added to the current position of each node, providing their estimated new position on the following time frame. Afterwards, this complete process was iterated over all the frames. Once all the frames were processed, the same procedure was applied in a backwards direction, i.e., starting with the nodes in their last computed positions and moving from the last frame to the first frame. Next, the weighted average between forward and backward tracking was computed for each frame, as described in [20], to provide one displacement matrix. Finally, this entire process was performed again after a modification of the initial node position to prevent bias from the initial node positioning on the final results. For the LCL, the nodes were translated ±0.5 mm along the x- and/or y-axis of the image, while for the thinner MCL, the displacement along the y-axis was only ±0.3 mm. By consequence, for each trial, nine displacements matrices, one per set of initial node position, were provided.

In the third phase, i.e., the post-tracking phase, for each trial, the nine displacement matrices obtained at the end of the tracking phase were averaged. Then, strains along the x-axis of the image *e_x,i_* were computed for each node *i* at each frame using the following small strain approximation [13]:(1)$$e_{x,i}=\frac{\Delta x\;\left(i-1,i+1\right)-\;\Delta x_{0}\;\left(i-1,i+1\right)}{\Delta x_{0}\;\left(i-1,i+1\right)}$$

To obtain the along-fiber strain values, i.e., aligned longitudinally with the ligament, the distances between each node were aligned with the longitudinal direction of the ligament at each frame using the tracked superficial border of the ligament as a reference orientation. Strains were then averaged over all nodes and linearly resampled to match a loading curve ranging from 0 to 40 N at 0.1-N intervals. Lastly, for each specimen, the overall strains were averaged over all trials. At the end of this third phase, an additional quality check was performed in terms of the tracking performance. First, trials with negative overall strains were discarded, since a negative strain is not representative of the physiologic behavior of a ligament under varus (LCL) or valgus (MCL) loading [21,22,23]. Therefore, these values were assumed to result from problematic tracking and/or bad assessment during the initial image quality assessment. Second, for each specimen, correlation coefficients during tracking were averaged over the entire length of the trials, over all nodes and over all valid trials. Then, specimens with an average value below 0.95, considered as a poor average tracking performance, were discarded.

#### 2.3.2. Digital Image Correlation

Cropped image sequences recorded by the high-resolution cameras of the DIC system were processed with commercial software VIC3D. The area surrounding the joint line, marked during specimen preparation, was selected at the approximate position of the US probe, i.e., around 50% of the ligament width, and processed with a subset and step size of 29 and 7 pixels, respectively. Data were transformed to have the x-axis of the strain map aligned along the longitudinal axis of the ligament, and the strains along the x-axis were extracted. The strains were then resampled at 0.1-N loading intervals and averaged over all trials for each specimen.

#### 2.3.3. Motion Capture

Following the segmentation of the CT scans (Mimics 18, Materialize NV, Leuven, Belgium) and identification of anatomical landmarks [24], subject-specific kinematic reference frames were defined for the femur and tibia of each leg. Then, motion capture data were time-cropped and processed using a custom pipeline (Nexus 2.9.2, Vicon Motion Systems) to obtain anatomical translations and rotations of the knee joint according to the Grood and Suntay convention [25]. Finally, the resulting kinematics were resampled at 0.1-N loading intervals and averaged over all trials of each ligament.

### 2.4. Analysis

#### 2.4.1. Experimental Comparability between DIC and US Trials

A linear mixed model with pairwise comparisons for load points (R-studio v1.0.143, Boston, MA, USA) was used to determine if the experimentally nonconstrained kinematics, i.e., add/abduction and internal tibial rotation angles, were significantly different (*p*-value < 0.05) between DIC and US acquisitions across all loading ranges. Average root mean square errors (RMSE) and linear correlation coefficients across the loading range were also calculated. Likewise, loading rates (LRs) between both DIC and US acquisitions were measured and compared with a nonparametric Wilcoxon test (*p*-value < 0.05).

#### 2.4.2. US Tracking Parameter Optimization

With the specimens from the “optimization” group, the following tracking parameters were selected to be optimized based on a prior work [5]: kernel and search window dimensions, simulated frame rate and minimum acceptable correlation coefficient during tracking. Initial values were based on data from a previous study [5]; then, the kernel and search window dimensions were adjusted through an iterative process where the results were ranked after each iteration step until no improvement was observed anymore, as illustrated in Figure 5. The ranking of the parameter sets was performed in three steps for each ligament type. First, for each specimen of the “optimization” group, a leave-one-out setup was used. Results were averaged over all the other specimens of the group, and tracking parameter sets were ranked based on their correlation coefficient between the DIC and US strains, average absolute strain differences over the loading range and absolute strain differences at the peak load of 40 N. Second, results were averaged over all specimens of the “optimization” group, and tracking parameter sets were ranked based on the same metrics. Third, for each tracking parameter set, the average rank over all optimization steps was computed for each ligament, and the set with the best average rank was selected as the “optimal ligament-specific tracking parameter set”.

#### 2.4.3. Tracking Performance

For each valid specimen of the “validation” group, both the LCL and MCL, strains were computed using the “ligament-type optimal tracking parameters set”; then, the DIC and US strains were compared. First, the correlation coefficients between the DIC and US strains were computed. Second, the mean absolute differences across the loading range and at 40 N were measured.

## 3. Results

### 3.1. Quality Assessment

First, based on the initial image quality assessment, no LCLs were removed, and two MCLs presenting important artefacts were discarded. Second, applying the two aforementioned quantitative criteria after strain computation using optimal parameter sets, additional specimens were removed from the analyses as follows. In total, five trials with negative overall strains were found, and all belonged to a single MCL specimen from the “validation” group that was discarded. In addition, an average correlation coefficient during tracking inferior to 0.95 was observed for one MCL specimen of the “validation” group and, thus, also removed before further analyses. A flowchart of the specimen evaluation can be found in Figure 6.

### 3.2. Experimental Comparability

Across the complete loading range, the only significant differences observed between the DIC and US acquisitions in terms of nonconstrained kinematics (Table 1) concern the rotation angles during MCL trials for a limited loading range from 12 to 15 N. In addition, the average RMSE were below 1° and correlation coefficients above 0.80 for both the LCL and MCL. In terms of LR, significant differences between the DIC and US trials were found for both the LCL and MCL, with *p*-values equal to 0.0175 and 0.0079, respectively. Higher LRs were observed for US acquisitions of LCL (LR_DIC_ = 25.32 ± 4.19 N/s and LR_US_ = 33.08 ± 4.97 N/s) and MCL (LR_DIC_ = 25.71 ± 3.89 N/s and LR_US_ = 34.59 ± 3.20 N/s).

### 3.3. Parameter Optimization

Following the optimization process, the best results were achieved with the parameters values summarized in Table 2. With these values, average correlation coefficients of 0.97 and 0.63 were observed for LCL and MCL, respectively, and average absolute differences over all specimens from the “optimization” group below 0.5% (Table 3).

### 3.4. Validation

During the validation process, correlation coefficients above 0.89 were observed for each individual specimen, as well as the absolute differences below 0.6% (Table 4).

## 4. Discussion

The aim of this study was to develop, optimize and validate an US speckle tracking approach to assess knee collateral ligament strains in situ with the potential to assist surgeons with knee balancing during TKA. The results obtained using the optimized ligament-specific tracking parameter set indicate that, for both the MCL and LCL, there is a strong correlation, together with acceptable absolute differences, between strains measured with our US speckle tracking method and reference DIC strains. These observations are in agreement with the results obtained during the validation of other processing pipelines based on US speckle tracking in tendons [13,26] or on ex-situ LCL [6]. In addition, the strain measurements obtained were accurate enough to detect the minimal strain deviation from pre- to post-TKA of 1.5% observed in a previous study [27], as well as the 1% of strain deviation [27] associated with frontal plane misalignment above 3° [28].

Although the internal tibial rotation was found to be significantly larger during US acquisitions of the MCL, this only occurred in a limited loading range from 12 to 15 N, and the RMSE was only 0.25 degrees. Except for this difference, no significant differences were observed between the DIC and US acquisitions in terms of nonconstrained kinematics, and strong correlations were measured. Therefore, differences observed during strain comparisons are not expected to be linked to the experimentally applied kinematics. Nonetheless, LRs were found to be significantly different, which could affect the strains measured, considering the viscoelastic nature of the collateral ligaments. Even though it was shown that different LRs lead to significant differences in terms of collateral ligament strains [29], the main differences were observed at high LRs and high ligament strains. Therefore, considering the small LRs and collateral ligament strains observed in this study, no impact on the measured strain was expected.

For the LCL, the results were consistent with results obtained on larger structures [4] and on LCL ex situ [6]. Indeed, the results obtained during optimization showed a high average correlation coefficient (above 0.90), as well as differences below 0.5% strain between DIC and US. In addition, these results obtained within the “optimization” group were supported by the results obtained on the two valid LCL specimens during the validation procedure (Table 4). For both specimens, the DIC and US curves (see Figure 7 for the DIC vs. US strain of individual specimens) followed similar strain-stiffening behaviors with correlation coefficients close to one and strain differences below 0.6%.

For the MCL, with a moderate average correlation of 0.63, the tracking performance seemed to be slightly reduced compared to LCL, despite similar mean absolute differences below 0.5%. Once again, these results obtained with the “optimization” group’s specimens were supported by the results obtained during validation (Table 4). Indeed, for both valid MCL specimens, the DIC and US curves followed a similar elongation pattern (see Figure 8 for the DIC vs. US strain of individual specimens), with correlation coefficients close to one and mean absolute differences even smaller than for the LCL with values below 0.3%. However, as illustrated by the MCL image quality assessments where four specimens had to be removed prior to analysis, it is more difficult to obtain images of the MCL with sufficient quality to allow a subsequent analysis. A first potential explanation for this is the fact that the MCL is much wider than the LCL [30]; hence, the ligament borders are not available as visual cues to help maintain a constant position of the probe with respect to the ligaments under investigation. Second, in contrast to the LCL, the MCL’s deformation is complicated by its wrapping around the femoral and tibial medial condyles, making it more difficult to follow and keep the probe in a consistent position during motion without losing contact with the ligament. Those two characteristics make it more difficult to obtain good initial quality images. Third, the MCL is thinner [30]; hence, fewer pixels constitute the ligament in the US image. Fourth, MCL strains are known to be much lower [23], which was also found in our “optimization” group demonstrating 1.48% vs. 2.92% strains for the MCL and LCL, respectively. This means that lower displacement-to-noise ratios likely impacted the MCL strain computation. These two additional parameters likely further contributed to the poorer tracking performances compared to the LCL. Nevertheless, the performance remained encouraging toward the intended clinical application and was consistent with the results previously reported on presumably easier-to-track structures [4].

Although a good overall tracking performance was achieved, we observed clearly higher within-specimen variability (standard deviations) in calculated US strains compared to the within-specimen strain variability based on DIC (Table 2 and Table 3). Indeed, the standard deviations reached up to 40% and 90% of the strains at 40 N for LCL and MCL, respectively. This observation highlights, similar to the previously published methods [2,5,6], the need to collect multiple trials, efficiently discarding bad ones and averaging the remaining trials to gain a correct final estimation of the in-situ collateral ligament strains. Even then, some US vs. DIC differences remain for both the LCL and MCL. However, as previously highlighted [6,13], these can be explained by multiple factors. First, the DIC data used as reference was measured on the surface of the ligament, while US data was obtained in the mid-portion of the ligament. Second, the DIC strains were based on a 3D deformation map, while, in the US, this was inherently limited to 2D deformations. Third, as mentioned in previous studies concerning US speckle tracking in collateral ligaments [5,6], the image quality remains an important issue, considering we tried to characterize a 3D phenomenon with a 2D modality. Fourth, the legs were implanted with a total knee prosthesis, hence displaying a potential situation in clinical application and thereby causing artefacts due to the reflection, as well as significantly modifying the acoustic impedance. Therefore, the differences between the DIC and US strains observed were expected but were mainly related to the differences in acquisition and processing.

In all US-based strain calculation methods, the main determinant of the tracking performance is the correct discarding of images with poor quality [5]. Unfortunately, no specific metric or methodology is reliable to quantify the initial image quality. In this study, similar to a previous study [5], trials with obvious low quality caused by probe motion, out-of-plane motion and metal artefacts were removed in a first subjective selection phase. Next, this work introduced the use of two other quantitative criteria to perform a second selection phase once the processing was completed. Firstly, trials with final negative strains—hence, not representative of physiologic behavior [21,22,23]—were excluded. Secondly, specimens with relatively low average correlation coefficients during tracking (<0.95) were discarded. These two criteria led to the removal of two MCLs from the “validation” group for which tracking performances were poor, probably due to subtle artefacts appearing during the complete loading phase that remained undetected by the ultrasonographer during the first visual assessment of the image quality. However, the second rule was based on the specimens available in this study. Indeed, the only valid specimen with an average correlation coefficient during tracking below 0.95 displayed poor final results compared to other specimens, with average coefficients all above 0.97. As a result, this threshold might be specific to this group of specimens, and further studies should be carried out to further optimize this quantitative criterion. Therefore, despite the new quantitative criteria introduced in this study, further works should be dedicated to the development of quantitative criteria and/or automatic trial discrimination methods, considering the importance of image quality assessment in US-based strain calculation methodologies.

The main limitation of this study is the low number of specimens. Indeed, this study was performed with only nine specimens (nine MCL and seven LCL), though this is a larger sample size than prior studies that used four or six specimens [4,6]. Additionally, the age range of the dataset exceeded the typical age of subjects slated for total knee arthroplasty (67.6 ± 10.2 years [31]). However, even though demographic parameters such as age can affect the absolute strains of soft tissues [32], they are not expected to affect the performed comparison of reference and ultrasound-based strains collected on same ligaments in similar experimental conditions. In addition, this study was performed on cadaveric legs with exposed ligaments, making it easier to locate the ligament of interest and enable the collection of ground truth data contrary to the in-vivo conditions. However, this makes it also more difficult to keep the probe stable and avoid slipping during acquisition. Finally, all specimens were implanted with a TKA, adding potential sources of error, such as the possible presence of metal artefacts and the absence of tissues between the ligament and the implant, hence modifying the acoustic impedance. Therefore, it is expected that the same protocol on native legs would show better results.

## 5. Conclusions

The present study is the first to develop, optimize and validate a US speckle tracking methodology that can successfully assess the MCL and LCL strains in situ in the presence of an implant and has potential to be translated to the clinic for TKA-related procedures. It should be noted that the tracking of the MCL is technically more complicated and slightly less successful compared to the LCL due to differences in terms of the morphology and biomechanical properties that complicate the acquisition and processing. In addition, this study highlighted the importance of the US image quality on the final results. Even though the results are encouraging, further work on the assessment of image quality should be carried out by developing quantitative criteria and/or automatic trial discrimination methodologies and acquisition. This could eventually allow further progress towards a clinical in-vivo application of US-based measurements of knee collateral ligament strains.

## Figures and Tables

**Figure 1 sensors-21-01895-f001:**
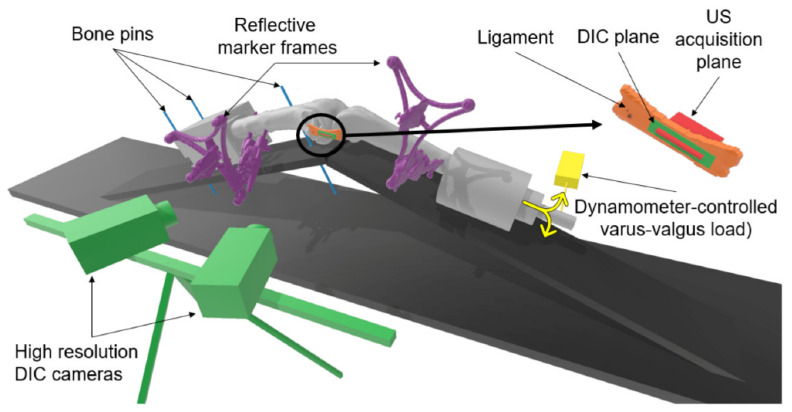
Experimental setup. Legs were positioned with a constant flexion–extension angle of 30°. The femur was rigidly attached using bone pins (**blue**), while the tibia was free to move and rotate relative to the femur to performed dynamometer-controlled varus–valgus loading (**yellow**). Rigid marker frames containing four reflective spheres each were pinned on the femur and the tibia (**purple**). High-resolution digital image correlation (DIC) cameras were placed to measure the strains on the ligament surface (**green**). An ultrasound probe was aligned in parallel with the longitudinal direction of the ligament (**orange**), centered on the joint line (**red**).

**Figure 2 sensors-21-01895-f002:**
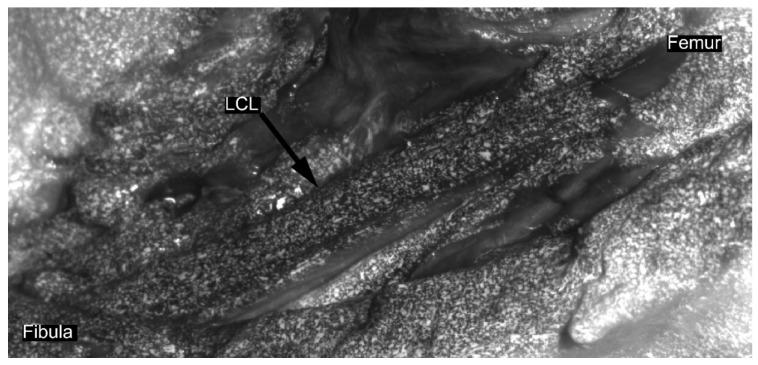
Representative speckle pattern for lateral collateral ligament prior strain computation using digital image correlation. First, diluted methylene blue was applied on the ligament to create a dark background; then, a speckle tracking was applied with white paint using an airbrush gun, following the previously published procedure [17].

**Figure 3 sensors-21-01895-f003:**
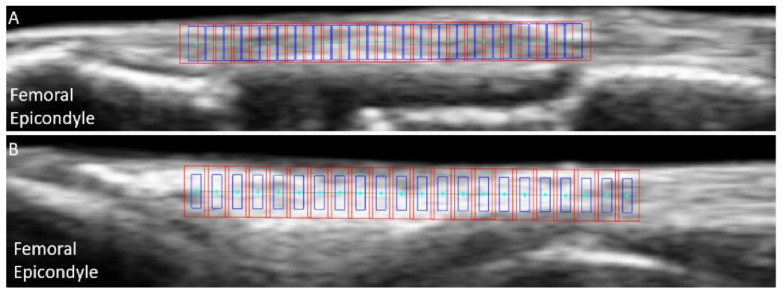
Example of initial nodes positions (cyan) overlaid on b-mode ultrasound images of the medial collateral ligament (**A**) and lateral collateral ligament (**B**), with the associated kernels (blue) and search windows (red).

**Figure 4 sensors-21-01895-f004:**
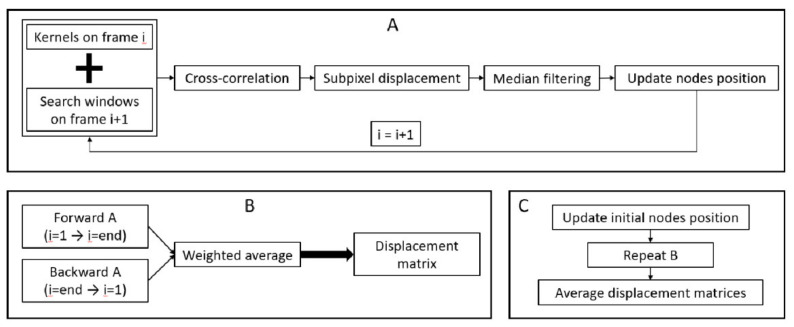
Schematic of the speckle tracking approach implemented for each trial. For each frame, the tracking of node positions is performed based on the normalized cross-correlation of the radiofrequency data (**A**). This tracking is performed in the forward and backward directions; after which, a weighted average is performed to obtain a displacement matrix (**B**). Finally, the initial nodes position is slightly modified (±0.5 mm along the x-axis of the image and ±0.5 or ±0.3 mm along the y-axis of the image for the lateral collateral ligament (LCL) and medial collateral ligament (MCL) respectively); the complete process is repeated, and all displacement matrices are averaged to obtain the final displacement matrix for the trial (**C**).

**Figure 5 sensors-21-01895-f005:**
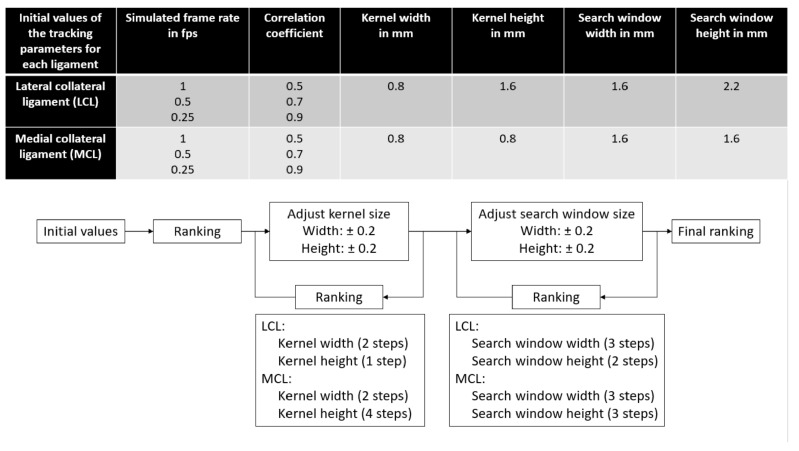
Tracking parameter values selection during optimization. Ultrasound data were first processed with initial values based on a previous study [5]. Afterwards, results were ranked, the dimensions of the kernel were modified around the best results and the ultrasound data were processed again. This processed was iterated until no improvement in the results were observed. Then, this whole iterative process was repeated similarly for the search window dimensions. Once this iterative process was over, a final ranking was performed to obtain the optimal parameter set for each ligament. fps: frames per second.

**Figure 6 sensors-21-01895-f006:**
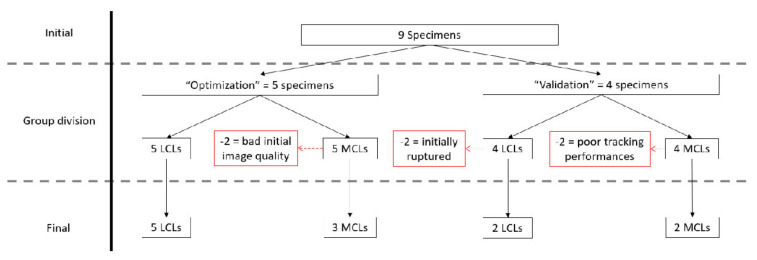
Flowchart of the specimen selection, from the initial population to the final valid specimens. Ligaments were discarded from the initial population (red) based on the initial conditions of the ligament, initial image quality analysis and tracking performances.

**Figure 7 sensors-21-01895-f007:**
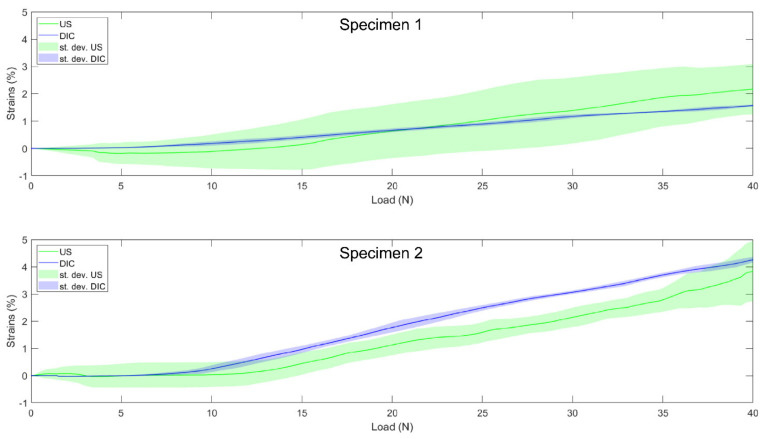
Individual load–strain curves for the two valid lateral collateral ligaments from the “validation” group, from 0 to 40 N. Solid lines represent the US (green) and DIC (blue) load–strain curves. Shaded areas represent the standard deviation across the trials of each specimen for US (green) and DIC (blue).

**Figure 8 sensors-21-01895-f008:**
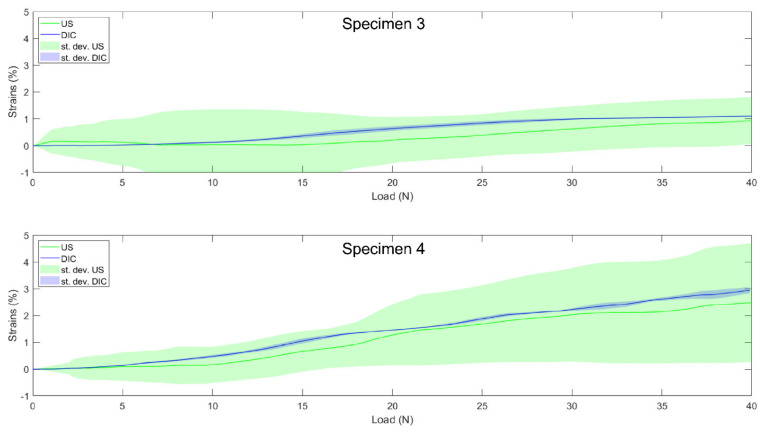
Individual load–strain curves for the two valid medial collateral ligaments from the “validation” group from 0 to 40 N. Solid lines represent the US (the green) and DIC (blue) load–strain curves. Shaded areas represent standard deviation across the trials of each specimen for US (green) and DIC (blue).

**Table 1 sensors-21-01895-t001:** Comparison of the unconstrained kinematics between the digital image correlation (DIC) and ultrasound (US) acquisitions. Average *p*-values across the loading range are reported with minimal and maximal values (*p*-values < 0.05 in red). Root mean square errors (RMSE) are expressed as the means across the loading range ± standard deviations in degrees. The linear correlation coefficient (R^2^) is reported as the average overloading range ± standard deviation. LCL: lateral collateral ligament and MCL: medial collateral ligament.

		*p*-Value (Average [min–max])	RMSE (Mean ± st. dev. in Degrees)	R^2^ (Mean ± st. dev.)
**LCL**	Adduction	0.48 (0.25–1)	0.65 ± 0.78	0.96 ± 0.06
	Rotation	0.40 (0.10–0.59)	0.21 ± 0.12	0.84 ± 0.14
**MCL**	Abduction	0.57 (0.10–0.99)	0.51 ± 0.30	0.98 ± 0.02
	Rotation	0.25 (0.04–0.65)	0.25 ± 0.08	0.98 ± 0.00

**Table 2 sensors-21-01895-t002:** Optimal values of kernel size (in mm), search window size (in mm), simulated frame rate (in frames per second) and acceptable correlation coefficient during tracking for both lateral collateral ligament (LCL) and medial collateral ligament (MCL).

	Kernel Size (Width × Depth in mm)	Search Window Size (Width × Depth in mm)	Simulated Framerate (fps)	Acceptable Correlation Coefficient
**LCL**	0.4 × 1.6	1 × 2.4	70	0.9
**MCL**	0.8 × 1.6	1.6 × 1.8	35	0.5

**Table 3 sensors-21-01895-t003:** Average results over all valid specimens from the “optimization” group obtained during the optimization process with the optimal parameter set of each ligament. Results of the linear correlation coefficient (R^2^) between the digital image correlation (DIC) and ultrasound (US) strains, and mean absolute difference over the entire loading range (in %) are expressed as average values over all specimens from the “optimization” group ± standard deviation. DIC and US strains at 40 N are reported as the average values over all specimens from the “optimization” group ± average within-specimen standard deviation in %.

	R^2^ (Mean ± std)	Average Absolute Difference (Mean ± st. dev. in %)	Average Strains at 40 N ± Average Strains st. dev. (%)
**LCL**	0.97 ± 0.3	0.37 ± 0.16	DIC: 2.92 ± 0.1
			US: 2.99 ± 1.16
**MCL**	0.63 ± 0.54	0.38 ± 0.11	DIC: 1.48 ± 0.06
			US: 1.65 ± 1.69

**Table 4 sensors-21-01895-t004:** Individual results of all valid specimens from the “validation” group processed with the optimal parameter set for each ligament. Results included the linear correlation coefficient (R^2^) between the digital image correlation (DIC) and ultrasound (US) strains, average absolute difference over all loading ranges (in %) and within-specimen average strains at 40 N for both the DIC and US ± standard deviation in %.

	Specimen	R^2^	Average Absolute Difference (in %)	DIC Strains at 40 N (Mean ± st. dev. in %)	Average US Strains at 40 N ± Average US Strains st. dev. (%)
**LCLs**	1	0.99	0.24	1.57 ± 0.03	2.17 ± 0.92
	2	0.99	0.57	4.26 ± 0.10	3.84 ± 1.11
**MCLs**	3	0.89	0.27	1.10 ± 0.02	0.94 ± 0.88
	4	0.99	0.26	2.94 ± 0.11	2.48 ± 2.21

## Data Availability

The data presented in this study are available on request from the corresponding author. The data are not publicly available due to ethical and privacy considerations associated with human cadaveric donor material.

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
