# Peer review of "Validated Ultrasound Speckle Tracking Method for Measuring Strains of Knee Collateral Ligaments In-Situ during Varus/Valgus Loading"

_sensors, 2021, doi:10.3390/s21051895_

Round 1

Reviewer 1 Report

Authors makes a proposal to perform a validated US speckle tracking method to perform meaurements of strains of Knee collateral ligaments. First, it must be posed that they only work with 9 cadaveric knee joints (all of them knee implants) from 5 male specimens, so the representation capabilities of the dataset can not be the best one specially if the age and the BMI of the are considered. Also, as they recognize at the paper two of them have broken LCLs and for the validation dataset the authors talk about “-2=poor tracking performances”. It is difficult to understand what was the problem at the tracking stage (2 cases in 4: 50%).

Moreover it is quite surprisingly that authors do not show any US image that they process at any stage, it would be useful to understand the process that support an US image based method.
But when they show the results at the discussion section the results seems to be worst. At Figure 4 Load-Strains curves for the LCL of the specimens are shown and at the lower figure US and DIC curves are separated with the std deviation included. Is it a “good” result?.

The procedures explanation must enhanced at most of their aspects: the three stages has not any figure to show how their procedures works.

Reviewer 2 Report

The manuscript was well prepared and the findings are very interestimg. Suggested minor changes and comments:

Page 1, title

measuring => Measuring

Page 3, L119, ‘a speckle pattern was applied on the ligament …’

Suggest adding a figure to show the speckle pattern. Does this pattern affect the tracking performance? Is it something possible in the clinical setting?

Page 3, L102-103, ‘… simultaneously collected and synchronized at 1000 Hz with LabView’

Why pick 1000 Hz as the synchronization frequency? How accurate is the synchronization? Is the trigger delay taken into consideration with?

Page 6, L225, ‘work5’

work [5]

Page 6, L227, Figure 2. caption, ‘study5’

study [5]

Page 6, Figure 2. ‘MCL: … Search window height ( steps)’

Missing number here

Page 7, Figure 3. blurred figure

Replace with the original figure

Page 11, L389, ‘…,  were’

Double space

Round 2

Reviewer 1 Report

Authors have performed most of changes that were proposed. In fact the dataset (that comes from a few cadaveric bodies) seems to be reduced but the consideration that it is quite difficult to get more instances (with artificial knee) can be accepted. Rest of the topics have been taken into account.

An enhanced explanation of Figure 2 must be provided.

Line 230. Initial values were based on data from a previous study [5]”,” then kernel

Line 260. mean absolute difference across the loading range “and at” 40 N” were measured
